# Long-Term Intake of Proton-Pump Inhibitors Could Be Associated with an Increased Incidence of Liver Cancer in Women

**DOI:** 10.3390/cancers16081517

**Published:** 2024-04-16

**Authors:** Sven H. Loosen, Markus S. Jördens, Catherine Leyh, Tom Luedde, Christoph Roderburg, Karel Kostev

**Affiliations:** 1Department of Gastroenterology, Hepatology and Infectious Diseases, University Hospital Düsseldorf, Medical Faculty, Heinrich Heine University Düsseldorf, 40225 Düsseldorf, Germany; markus.joerdens@med.uni-duesseldorf.de (M.S.J.); catherine.leyh@med.uni-duesseldorf.de (C.L.); tom.luedde@med.uni-duesseldorf.de (T.L.); christoph.roderburg@med.uni-duesseldorf.de (C.R.); 2Epidemiology, IQVIA, 60549 Frankfurt, Germany; karel.kostev@iqvia.com

**Keywords:** PPI, HCC, CCA, malignancy, correlation, epidemiology

## Abstract

**Simple Summary:**

Proton pump inhibitors (PPIs) are very commonly prescribed drugs in gastroenterology. The present study evaluated a potential association between PPI intake and a subsequent diagnosis of liver cancer in Germany. We observed that long-term PPI intake in women as well as in patients < 60 years might be associated with an increased risk of liver cancer. These findings support current efforts to reduce the inappropriate use of PPIs in clinical routine.

**Abstract:**

Background: Proton pump inhibitors (PPIs) are among the most commonly prescribed drugs in gastroenterology. Although PPIs are mostly well tolerated, long-term PPI intake has been linked with diabetes mellitus, osteoporosis and infectious disease. In the present study, we evaluated a potential association between PPI intake and a subsequent diagnosis of liver cancer in a large real-world cohort of outpatients in Germany. Methods: A total of 1766 patients with liver cancer, as well as 8830 propensity-score-matched controls, were identified from the Disease Analyzer database (IQVIA). The outcome of the study was the association between PPI use and a subsequent diagnosis of liver cancer, which was evaluated using multivariable logistic regression analyses. Results: Overall, 42.9% of the liver cancer patients and 39.0% of the controls received at least one PPI prescription before the index date. PPI prescriptions at any time before the index date were associated with an increased risk of subsequent liver cancer (OR: 1.18; 95% CI: 1.06–1.31). The positive association was observed in all age groups, as well as in women and men, but only in women (OR: 1.30; 95% 1.09–1.55) did it reach the predefined level of significance (*p* < 0.01). When considering the duration of PPI therapy, only PPI therapy for at least two years was significantly associated with an increased risk of liver cancer (OR: 1.28; 95% 1.09–1.50). In an analysis stratified by age and sex, this association was strongest in the age group < 60 years (OR: 1.99; 95% 1.21–3.26). Conclusions: Our data suggest that long-term PPI intake in women as well as in patients < 60 years might be associated with an increased risk of liver cancer. These findings support current efforts to reduce the inappropriate use of PPIs in routine clinical practice and to link PPI prescribing to a clear medical indication.

## 1. Introduction

Proton pump inhibitors (PPIs) play a decisive role in the clinical management of various gastrointestinal conditions by suppressing stomach acid production. Widely prescribed for conditions such as peptic ulcers or gastroesophageal reflux disease (GERD), PPIs have become a mainstay in the treatment of these acid-related disorders [1]. While their efficacy for several medical indications is well-established, the long-term use of PPIs has raised concerns regarding potential negative sequela. For example, the prolonged use of PPIs has been associated with a reduced absorption of certain nutrients, such as magnesium [2,3], calcium [4] or vitamin B12 [5]. This can lead to complications such as osteoporosis, with an increased risk of fractures [6,7,8]. Moreover, PPIs, by suppressing gastric acid production, may increase the risk of gastrointestinal infections, such as *Clostridium difficile* (*C. difficile*) infections [9]. Some studies have also suggested a potential link between long-term PPI use and an increased risk of kidney disease as well as cardiovascular events, though the causal relationship remains a topic of investigation [10,11,12].

The association between proton pump inhibitors (PPIs) and an increased risk of cancer is a topic of ongoing research and debate. While some studies have suggested potential links between long-term PPI use and certain types of cancer, the evidence is not conclusive, and more research is needed to establish a clear connection [13]. Specifically, some studies have explored the potential association between PPI use and an increased risk of gastric cancer, as well as concerns about the development of adenocarcinoma in the esophagus [14,15,16]. However, the findings have been mixed, and it is challenging to establish a direct cause-and-effect relationship.

To date, a potential association between PPI intake and the development of liver cancer is only poorly understood. While some studies suggest an increased risk of liver cancer and mortality in patients with chronic liver disease [17], there are currently no systematic data from Germany regarding the risk of liver cancer development as a sequela of PPI intake for the general population. Therefore, in the present study, we aim to investigate the association between PPI intake and a subsequent diagnosis of liver cancer in a large real-world cohort of outpatient treated in a general practice setting in Germany. 

## 2. Materials and Methods

### 2.1. Database

This study used data from the Disease Analyzer database (IQVIA). Details of this database have been published previously [18]. In brief, the Disease Analyzer database contains data on demographic variables, diagnoses and prescriptions of outpatients from general practices in Germany. The database covers about 1300 general practices in Germany. The panel of practices included in the Disease Analyzer database has previously been shown to be representative of office-based practices in Germany [18]. The database has been widely used for epidemiological studies in recent years, including studies on PPIs [9,19] or cancer [20,21].

### 2.2. Study Population

The study population included all patients aged ≥18 years with a first diagnosis of liver cancer (index date, ICD-10 codes: C22) between January 2015 and December 2022 (index date) who had at least 1 year of follow-up before the index date. The controls were individuals with no history of cancer who were matched (1:5) by nearest neighbor propensity scores based on age, sex and diagnoses documented before the index date, including diabetes mellitus (ICD-10: E10–E149), obesity (ICD-10: E66), chronic obstructive pulmonary disease (COPD) or chronic bronchitis (ICD-10: J42–J44), esophageal and gastric diseases (ICD-10: K20–K31), and liver diseases including non-alcoholic fatty liver disease (NAFLD) (ICD-10: N75. 8, N76.0), chronic hepatitis (ICD-10: B18, K70.1, K75.2, K75.3, K75.4) and cirrhosis (ICD-10: K70.2, K70.3, K74). For individuals without cancer, the index date was a randomly selected visit date between January 2015 and December 2022. The flow diagram of the study participants is shown in Figure 1.

### 2.3. Study Outcomes

The outcome of the study was an association between PPI use documented before the index date and a subsequent liver cancer diagnosis. We included all PPI prescriptions in the complete patient history up to six months before the index date.

### 2.4. Statistical Analyses

Statistical analyses were carried out as previously described [22]. The demographic and clinical characteristics of the cases and controls after 1:5 propensity score matching were evaluated using the Wilcoxon signed-rank test for continuous variables, the McNemar test for categorical variables with two categories and the Stuart–Maxwell test for categorical variables with more than two categories. To examine whether PPI prescriptions were associated with subsequent liver cancer diagnoses, we used univariable logistic regression models and estimated odds ratios (ORs) with 95% confidence intervals (95% CIs). Two models were calculated. The first model included ever versus never PPI use, because the second model included PPIs categorized by duration of therapy (duration of PPI therapy ≤ 3 months, >3 months—≤1 year, >1 year—≤2 years and >2 years). Both models were also calculated separately for four age groups (<60 years, 60–69 years, 70–79 years, 80+ years) and for female and male patients. For multiple comparisons, a *p*-value < 0.01 was considered statistically significant. All the analyses were performed using SAS version 9.4 (SAS Institute, Cary, NC, USA).

## 3. Results

### 3.1. Basic Characteristics of the Study Sample

After 1:5 matching, 1766 cases (patients with liver cancer) and 8830 controls (patients without cancer) were available for analyses. The mean (standard deviation, SD) age at the index date was 71.0–71.1 (11.8) years. In total, 67% of the individuals were female. On average, both the cases and controls had an observation period of 10 years before the index date. The prevalence of the most common co-diagnoses was similar in the cases and controls (Table 1).

### 3.2. Association between PPI Use and Subsequent Liver Cancer Diagnosis

In total, 42.9% of the cases and 39.0% of the controls received at least one PPI prescription before the index date. PPI prescriptions at any time before the index date were associated with an increased risk of subsequent liver cancer (OR: 1.18; 95% CI: 1.06–1.31, Table 2). The positive association was observed in all age groups, as well as in women and men, but only in women (OR: 1.30; 95% 1.09–1.55) did it reach the predefined level of significance (*p* < 0.01). In the age groups < 60 years (OR: 1.33; 95% CI: 1.02–1.74) and 80+ years (OR: 1.25; 95% CI: 1.03–1.52), there was a trend towards an association with a *p*-value < 0.05 but >0.01 (Table 2).

### 3.3. The Role of PPI Therapy Duration

When considering the duration of PPI therapy, only PPI therapy for at least two years was significantly associated with an increased risk of liver cancer (OR: 1.28; 95% 1.09–1.50, Table 3). In an analysis stratified by age and sex, this association was strongest and significant only in the age group < 60 years (OR: 1.99; 95% 1.21–3.26, Table 3).

## 4. Discussion

Proton pump inhibitors (PPIs) have emerged as a cornerstone medication in the management of various gastrointestinal disorders by suppressing gastric acid production. While PPIs provide effective relief from acid-related symptoms, their widespread and often prolonged use has prompted investigations into potential sequelae, raising concerns about associated health risks. In the present study, we evaluated a potential association between PPI intake and a subsequent diagnosis of liver cancer. We were able to show that PPI therapy for at least two years was significantly associated with an increased risk of liver cancer, especially in the age group below 60 years.

A potential association between PPI intake and an increased risk of liver cancer has been the subject of scientific endeavor for some time, but reliable data either clearly establishing or ruling out this association are missing [23]. In a systematic review and meta-analysis, Song and co-workers evaluated a collective of 173,894 patients and observed that individuals with chronic liver disease (CLD) who used PPIs had a 67% greater risk of developing hepatocellular carcinoma (HCC) compared to nonusers. In addition, they described a 57% increased risk of mortality in PPI users with CLD compared to CLD patients who were nonusers [17]. The authors hypothesized that the positive correlation between PPI intake and liver cancer development was mainly due to the fact that all the patients had CLD, representing an established risk-factor for HCC development. Importantly, in our study collective, the proportion of patients with liver cirrhosis was well balanced between liver cancer and non-liver cancer patients (*p* = 0.893), suggesting that the association between PPI and liver cancer development is at least partially independent of pre-existing CLD. In another nested case-control study in a cohort of patients without viral hepatitis in Taiwan including 29,473 HCC cases and 294,508 matched controls, the adjusted odds ratio (AOR) for HCC associated with PPI use was 2.86. Interestingly, the authors observed a higher AOR in patients who had received a higher cumulative defined daily dose, arguing for a dose-dependent association between PPI intake and the development of liver cancer [24]. In our collective of patients, we also observed an increasing OR with respect to liver cancer development in patients with long-term PPI intake. The strongest, and only significant, OR (1.28) was found in patients with a cumulative PPI intake of more than two years. In a second meta-analysis of cohort studies, Tran and co-workers likewise described a positive correlation between PPI intake and liver cancer development, with an OR of 1.55 [13]. Interestingly, they also observed an increased risk of gastrointestinal cancer (the study also described a positive association with esophageal, gastric and pancreatic cancer) in individuals who had used PPIs for less than 1 year [13]. In comparison to our data, the OR for liver cancer development following PPI intake of the beforementioned studies is higher, which is most likely due to the heterogeneity of study designs including factors such as duration of PPI intake and study inclusion criteria (e.g., patients with or without CLD) as well as differences in liver cancer diagnosis and screening. 

In contrast, other studies did not find a significant association between PPI intake and the subsequent development of liver cancer. For example, in a large cohort study including 35,356 patients with chronic hepatitis B or C virus (HBV/HCV) infections from the Taiwan National Health Insurance Research Database, PPI use was not associated with the risk of developing HCC. In the HBV/HCV cohort, 237/211 patients developed liver cancer during a median follow-up of 53 months. However, PPI use was not associated with an increased risk of developing liver cancer HCC (*p* = 0.18/0.25) [25]. In a different meta-analysis, Chang and colleagues showed that PPIs were associated with HCC (crude risk ratio: 2.27; *p* < 0.01) when an unadjusted risk ratio was adopted. However, when they used data that were adjusted only by comorbidities and concurrent medications, the association between PPIs and HCC became insignificant (adjusted RR: 1.62; *p* = 0.11) [26]. These rather conflicting results nicely illustrate the scope of the ongoing debate and underline the need for larger, systematic studies to evaluate the clinically very important hypothesis that (long-term) PPI intake is associated with an increased incidence of liver cancer.

Our study also provided evidence for a sex-specific association between long-term PPI intake and liver cancer development. In the subgroup analysis, the OR for liver cancer development was significant in female patients only. Although liver cancer is more prevalent among male patients [27], this finding argues for a sex-specific effect of PPI intake in terms of liver cancer development. Interestingly, there is a growing body of evidence suggesting sex differences in the pathogenesis of liver cancer [28], and several cytokines as well as sex hormones have been suggested as the underlying pathophysiological basis [29,30]. The extent to which PPI influences these gender-specific pathophysiological mechanisms has not yet been sufficiently investigated [31,32]. We therefore encourage further translational and molecular analyses to investigate the potential link between PPI use and liver cancer. 

Importantly, our study is merely observational and cannot provide any causal relationship between PPI intake and the development of liver cancer in male or female patients. Several hypotheses have been proposed to explain the association between PPI intake and liver cancer development. PPIs have been shown to alter the composition and diversity of the gut microbiota, a condition that can lead to hepatic inflammation and liver cancer development [33,34]. Moreover, PPI intake can lead to an increase in gastrin levels due to the removal of the negative feedback inhibition of gastrin release. Elevated levels of gastrin have been associated with pro-oncogenic characteristics such as tumor growth [35,36].

Our study is limited by certain aspects. First, in an outpatient clinical setting, we acknowledge that some of the diagnoses may be subject to incorrect coding or misclassification by the attending physician. There is no available information with respect to the methods of liver cancer diagnosis, and often diagnoses recorded by the general practitioner have been previously established by a liver specialist or in a hospital setting without the GP’s involvement. Second, the Disease Analyzer database used for our analysis does not provide systematic data on individual factors such as the socioeconomic status of patients (e.g., education and income) or lifestyle-related risk factors (e.g., smoking, alcohol consumption and physical activity) as well as medication intake which may potentially be associated with a bias. This also applies to various laboratory parameters including AST and ALT levels, which have only been documented in a small minority of included patients. Thirdly, low-dose PPIs (e.g., pantoprazole 20 mg) can be bought over the counter without consulting a GP. Therefore, we cannot exclude the possibility that individual patients classified as non-PPI patients were taking low-dose PPIs without the knowledge and documentation of their GP. Finally, it is important to note that our study design only allows us to hypothesize the associations between the variables and not to infer causality.

## Figures and Tables

**Figure 1 cancers-16-01517-f001:**
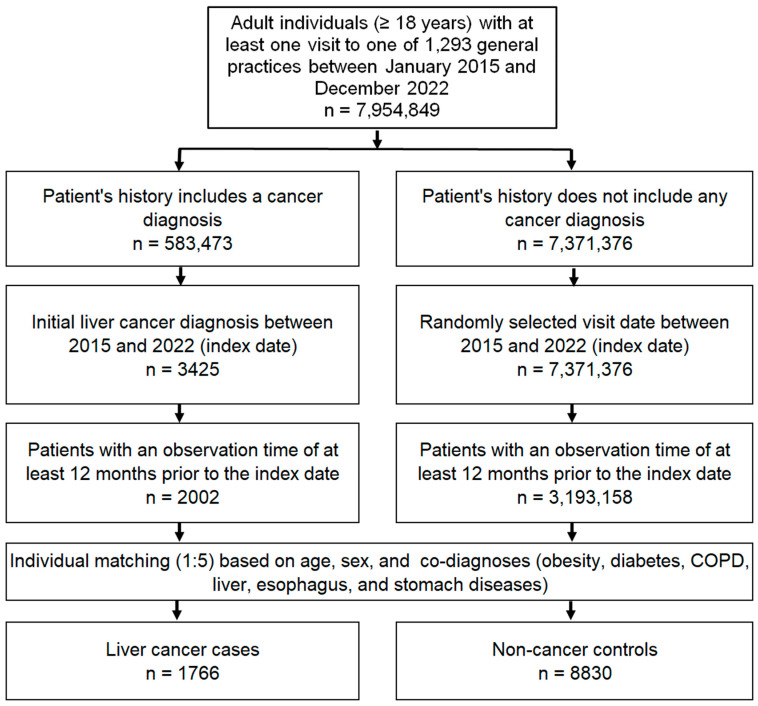
Selection of study patients.

**Table 1 cancers-16-01517-t001:** Characteristics of study patients after 1:5 matching.

Variable	Liver Cancer (*n* = 1766)	No Cancer (*n* = 8830)	*p*-Value
Age (in years)			
Mean (SD)	71.0 (11.8)	71.1 (11.8)	0.767
<60	282 (16.0)	1397 (15.8)	0.996
60–69	465 (25.8)	2266 (25.7)	
70–79	550 (31.1)	2770 (31.4)	
80+	478 (27.1)	2397 (27.1)	
Sex			
Female	583 (33.0)	2922 (33.1)	0.949
Male	1183 (67.0)	5908 (66.9)	
Observation time prior to the index date (years), mean (SD)	9.8 (6.7)	9.8 (7.4)	0.149
Conditions documented prior to index date			
Diabetes mellitus	830 (47.0)	4287 (48.7)	0.234
Obesity	299 (16.9)	1513 (17.1)	0.836
COPD/chronic bronchitis	286 (16.2)	1451 (16.4)	0.805
Esophagus/stomach diseases	606 (34.3)	3029 (34.3)	0.993
NAFLD	195 (11.0)	959 (10.9)	0.823
Chronic hepatitis	43 (2.4)	146 (1.8)	0.059
Liver cirrhosis	211 (12.0)	1045 (11.8)	0.893

Data are absolute samples and percentages unless otherwise specified. SD = standard deviation.

**Table 2 cancers-16-01517-t002:** Association between PPI prescriptions (ever used vs. never used) and subsequent liver cancer diagnosis in patients followed in general practices in Germany.

Variable	Proportion of Patients with PPI Prescriptions among Liver Cancer Cases (%)	Proportion of Patients with PPI Prescriptions among Controls (%)	OR (95% CI)
All patients	42.9	39.0	1.18 (1.06–1.31) *
<60 years	37.9	31.5	1.33 (1.02–1.74)
60–69 years	40.4	37.9	1.11 (0.90–1.36)
70–79 years	43.1	40.4	1.12 (0.93–1.34)
80+ years	48.1	42.6	1.25 (1.03–1.52)
Women	44.1	37.8	1.30 (1.09–1.55) *
Men	42.4	39.5	1.12 (0.99–1.28)

Abbreviations: OR, odds ratio; CI, confidence interval. * *p* < 0.01.

**Table 3 cancers-16-01517-t003:** Association between PPI prescriptions (depending on therapy duration) and subsequent liver cancer diagnosis in patients followed in general practices in Germany.

	Odds Ratios (OR) (95% CI)						
Variable	All Patients	<60 Years	60–69 Years	70–79 Years	80 + Years	Women	Men
No PPI	Reference	Reference	Reference	Reference	Reference	Reference	Reference
PPI therapy duration ≤3 months	1.17 (0.99–1.39)	1.32 (0.90–1.93)	1.19 (0.85–1.65)	1.05 (0.76–1.46)	1.19 (0.84–1.69)	1.17 (0.86–1.60)	1.17 (0.95–1.44)
PPI therapy duration >3 months—≤1 year	1.08 (0.92–1.27)	1.20 (0.81–1.80)	1.09 (0.79–1.49)	0.94 (0.70–1.26)	1.18 (0.88–1.59)	1.28 (0.98–1.68)	0.99 (0.81–1.21)
PPI therapy duration >1 year—≤2 years	1.20 (0.98–1.48)	0.89 (0.42–1.98)	1.11 (0.71–1.73)	1.18 (0.83–1.67)	1.39 (0.97–2.00)	1.35 (0.94–1.94)	1.13 (0.88–1.47)
PPI therapy duration >2 years	1.28 (1.09–1.50) *	1.99 (1.21–3.26) *	1.07 (0.77–1.48)	1.33 (1.01–1.74)	1.27 (0.95–1.69)	1.39 (1.06–1.84)	1.23 (1.01–1.50)

Abbreviations: OR, odds ratio; CI, confidence interval. * *p* < 0.01.

## Data Availability

The underlying data are available upon reasonable request from the corresponding author.

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
