# Peer review of "Long-Term Intake of Proton-Pump Inhibitors Could Be Associated with an Increased Incidence of Liver Cancer in Women"

_cancers, 2024, doi:10.3390/cancers16081517_

Round 1

Reviewer 1 Report

Comments and Suggestions for Authors

Comments:

1. Can authors explain or discuss why long-term PPI use is not associated with liver cancer development in men?

2. Since the study concludes that long-term PPI use is associated with women, the title will be better to add " liver cancer in women".

3. On Table 1, more info is needed: smoking history, alcohol use, BMI, any diarrhea, any constipation, history of bicarbonate use, H. pylori history, AST level, ALT level, and AFP level?

4. Since it is associated with liver cancer, AST, ALT, and AFP levels are required.

Author Response

We would like to thank this referee for her/his positive and constructive comments on our manuscript. We have now addressed all points raised by this reviewer and were able to improve our manuscript in quality and focus. We now hope that our manuscript will be deemed suitable for publication.

  1. Can authors explain or discuss why long-term PPI use is not associated with liver cancer development in men?

We thank the reviewer for this very relevant comment. Although our observational study is not able to answer this decisive question why long-term PPI use is associated with liver in cancer in females only, we have now added a respective section on this relevant issue in the revised discussion section of our manuscript:

“Our study also provided evidence for a sex-specific association between long-term PPI intake and liver cancer development. In the subgroup analysis, the OR for liver cancer development was significant in female patients only. Although liver cancer is more prevalent among male patients [29], this finding argues for a sex-specific effect of PPI intake in terms of liver cancer development. Interestingly, there is a growing body of evidence suggesting sex differences in the pathogenesis of liver cancer [30], and several cytokines as well as sex hormones have been suggested as the underlying pathophysiological basis [31,32]. The extent to which PPI influences these gender-specific pathophysiological mechanisms has not yet been sufficiently investigated [33,34]. We therefore encourage further translational and molecular analyses to investigate the potential link between PPI use and liver cancer. Importantly, our study is merely observational and cannot provide any causal relationship between PPI intake and the development of liver cancer in male or female patients.” (pages 6 and 7 of the revised manuscript)

  1. Since the study concludes that long-term PPI use is associated with women, the title will be better to add " liver cancer in women".

We thank the reviewer for this relevant comment. We have amended the title accordingly.

  1. On Table 1, more info is needed: smoking history, alcohol use, BMI, any diarrhea, any constipation, history of bicarbonate use, H. pylori history, AST level, ALT level, and AFP level?

We would like to thank this reviewer for this important comment. We fully agree that Table 1 and our study in general would benefit from more detailed patient characteristics including smoking history, alcohol use, bowel habits, medication or various laboratory parameters. Unfortunately, the depth of data in the Disease Analyzer database is limited and we are unable to get this information for the majority of patients. This is a disadvantage of large-scale data analysis. While the enormous number of patient cases enables statistically reliable results, the lack of detail in the data often limits further desirable analyses. We have highlighted this important limitation in the revised discussion section as follows:

“Second, the Disease Analyzer database used for our analysis, does not provide systematic data on individual factors such as the socioeconomic status of patients (e.g., education and income) or lifestyle-related risk factors (e.g., smoking, alcohol consumption and physical activity) as well as medication intake which may potentially be associated with a bias.” (page 7 of the revised manuscript)

  1. Since it is associated with liver cancer, AST, ALT, and AFP levels are required.

We thank the referee for this relevant comment. As stated before, we unfortunately cannot provide these relevant laboratory values for many patients. Specific lab values such as AFP is usually not determined by GPs in Germany. Therefore, the DA database does not include this information. We double checked the database and found AST/ALT values for less than 5% of the study population. Therefore, we believe that no statistically reliable analysis is possible with this subgroup of patient. We have also included this relevant limitation to the revised discussion section of our manuscript.

“This also applies to various laboratory parameters including AST and ALT levels, which have only been documented in a small minority of included patients.” (page 7 of the revised manuscript)

Reviewer 2 Report

Comments and Suggestions for Authors

Topic of manuscript and quality is suitable for the cancers. I clearly see the size of the patient file as a strength of the manuscript.

Could be authors more specific if the risk of PPIs is related to their primary therapeutic effect or their side effect?

Author Response

Topic of manuscript and quality is suitable for the cancers. I clearly see the size of the patient file as a strength of the manuscript.

We would like to thank this referee for her/his positive and constructive comments on our manuscript. We have now addressed all points raised by this reviewer and were able to improve our manuscript in quality and focus. We now hope that our manuscript will be deemed suitable for publication.

Could be authors more specific if the risk of PPIs is related to their primary therapeutic effect or their side effect?

We thank the reviewer for this very important remark. This is indeed a very interesting question that we would love to answer precisely. However, our study is merely observational and cannot provide any causal relationship between PPI intake and the development of liver cancer in male or female patients. In the literature, several hypotheses have been proposed to explain the association between PPI intake and liver cancer development. PPIs were shown to alter the composition and diversity of the gut microbiota, a condition that could lead to hepatic inflammation and liver cancer development. Moreover, PPI intake can lead to an increase in gastrin levels due to the removal of the negative feedback inhibition on gastrin release. Elevated levels of gas-trin have been associated with pro-oncogenic characteristics such as tumor growth. To fully comply with this reviewer’s comment, we have discussed this very relevant issue in the revised discussion section of our manuscript:

“Importantly, our study is merely observational and cannot provide any causal relationship between PPI intake and the development of liver cancer in male or female patients. Several hypotheses have been proposed to explain the association between PPI in-take and liver cancer development. PPIs were shown to alter the composition and diversity of the gut microbiota, a condition that could lead to hepatic inflammation and liver cancer development [35,36]. Moreover, PPI intake can lead to an increase in gastrin levels due to the removal of the negative feedback inhibition on gastrin release. Elevated levels of gas-trin have been associated with pro-oncogenic characteristics such as tumor growth [37,38].” (page 7 of the revised manuscript)

Round 2

Reviewer 1 Report

Comments and Suggestions for Authors

No more comments